# Investigation on Aesthetic and Water Permeability of Surface Protective Material under Accelerated Weathering

**DOI:** 10.3390/ma14226896

**Published:** 2021-11-15

**Authors:** Feng Chen, Nguyen Xuan Quy, Jihoon Kim, Yukio Hama

**Affiliations:** 1Division of Engineering, Muroran Institute of Technology, Muroran 050-8585, Japan; cf645573067@gmail.com; 2Department of Civil Engineering, Hanoi Architectural University, Hanoi 100000, Vietnam; egcmat@gmail.com; 3College of Environmental Technology, Muroran Institute of Technology, Muroran 050-8585, Japan; bmjhun@mmm.muroran-it.ac.jp

**Keywords:** surface protective material, aesthetic, water permeability, accelerated weathering, durability, anti-soiling test, appearance, contact angle

## Abstract

In this paper, experiments were conducted on the effects of aesthetic and durability of three representative surface protective material under accelerated weathering test for 5000 h. First, the adaptability of the surface protective material coating to the substrate was proven by examining the aesthetic properties and the water permeability of the building materials. Second, the pollutant resistance of the surface protective material coating to artificial stain was assessed using xenon-arc light. The result shows that the appearance of the silane types did not change significantly, and the water permeability was improved. In addition, the silicate types did not improve water permeability and the surface color was changed. Fluor- resin types effectively improved the water permeability, but the surface color became dark. Sample measurements showed changes in the average width of the contamination after weathering, with an increase after spray cleaning and ultrasonic cleaning. However, it was observed that after washing the pollution average width of all specimens due to weathering at 5000 h was almost as much or smaller than the initial value.

## 1. Introduction

Concrete is widely used for building constructions because its convenient construction, low cost and easy usefulness of raw materials. Without the protection of exterior decoration materials, architectural concrete is prone to be influenced by different factors during service. It may be vandalized by UV radiation, lower temperature or cyclic wetting–drying. All those adversities mentioned above will not only threaten the durability of architectural concrete structures but damage its shiny surface and defile its beauty [1,2,3,4].

Tile finishes have been commonly used in the external walls of buildings since the 1970s because it is an extremely durable, aesthetic, and water-resistant material that is perfect for exterior walls. It is used commonly in high-rise buildings in Japan. However, because of the earthquake and weather factors, including temperature, humidity, wind, sunshine, and air pressure [5], the delamination of the exterior wall tiles has caused many casualties and injuries in the past decade. In addition, it is very difficult to maintain and inspect the outer walls of high-rise buildings, which require manual labor such as ropes or baskets. Both the aesthetic and durability of concrete facades can be enhanced by coatings. In recent years, the surface protective material is increasingly used. Because the surface protective material has the advantages like the easy construction, relatively cheap and lower industrial waste when repair, and so on.

European standard EN 1504 defines three types of surface treatments [6]: silane and siloxane-based (water repellent) coatings, silicate-based (pore blockers, also known as water glass) impregnations, and coatings that create a continuous protective layer along the concrete surface. The first type of coating created a hydrophobic layer on the exposed concrete surface. The impregnation reaction product can clog the pores partially or completely filling the capillary pores. In the first type of impregnation, the active ingredient product creates a thin hydrophobic layer on the pores, while in the second type, the reaction product can clog the pores and strengthen the concrete surface.

However, surface coating materials are a material that, like many others, degrade over time once put into use [7,8]. Ivanovna [9] and Garrido et al. [10] indicated that the moisture is absorbed by the coating and a change in the molecular structure occurs along with disruption of the pigment–film relationship, which leads to the initial stage of the destruction of the coatings, resulting in a change in color (the appearance of whiteness) and a loss of shine. The surface coatings tend to degrade over time under exposure to outdoor fluctuating conditions of temperature, humidity and ultra-violet (UV) radiation [11,12,13]. When silanes/siloxanes are exposed to elevated temperatures or UV aging, they become less effective in reducing water absorption. Some anomalies (e.g., efflorescence, discoloration) arise from a weathering process that is induced by environmental influences such as temperature, solar radiation and rain. However, the weathering effect is widely considered a major contributor to the degradation phenomenon of exterior wall protective coatings, but there are very few studies of the relationships between them.

In addition, the inadequate selection of materials or design/execution errors can lead to the degradation of surface coating materials. Thus, in order to prevent premature deterioration, surface coating materials must be appropriately prescribed, considering the factors that most affect their degradation.

L. Basheer [14] and P.A.M. Basheer et al. [15] demonstrated that the reduction in water absorption due to the application of pore-liners was effectively measured using Autoclam. Moradllo et al. [16] evaluated the time-dependent performance of concrete surface coatings in tidal zone of marine environment. In that study, surface coatings appropriately decreased chloride diffusion coefficient in first stage of exposure (up to 9 months), but the decreased rate of diffusion coefficient of surface coatings was less than reference specimen in the exposure times more than 9 months. This was specifically because of gradual deterioration of surface coatings. Furthermore, numerous studies have investigated and confirmed the benefit of silane-based pore liners on the durability of concrete [17,18,19,20]. Additionally, sealing concrete with soluble sodium silicate may improve surface properties such as hardness, permeability, chemical durability, and abrasion resistance [21]. 

In relation to the aging conditions which the specimens treated, surface coatings degrade when exposed to fluctuating conditions of temperature, humidity, and ultraviolet (UV) radiation. At elevated temperatures or UV aging, silanes or siloxanes become less effective in reducing water absorption. From an early age up to a year, some of these coatings can be quite effective in preventing degradation, but after that, their effectiveness may gradually decrease. However, according to other studies, the residual effect of such coatings still results in better service life for treating concrete [7,22]. Graziani et al. [23], assessed the durability of the hydrophilic nature and self-cleaning ability of TiO_2_nano coatings applied to a fired clay brick substrate using ultraviolet (UV) lamps and simultaneous UV and wetting/drying cycles. The authors concluded that the remaining TiO_2_ nanoparticles, although showing some surface degradation, kept practically the same efficiency as before the tests.

A preliminary study conducted by Grüllet et al. [24] showed that the opaque coating systems had not reached the limit state after 60 weeks of artificial weathering. The opaque paints can last up to 10 years of outdoor exposure as described by Williams et al. [25]. In a study by to Marsich et al. [26], the effect of artificial weathering on PP coextruded tape and laminate Indiana limestone that was heated at 100 °C, 200 °C, 300 °C, 400 °C and 500 °C for 1 h, 4 h and 16 h was evaluated to understand the effect of heating as an artificial weathering method for stone. A decrease in dynamic elastic modulus linearly proportional to the heating temperature was found for the heated samples. Some authors [27,28] studied artificial weathering of stone by heating. The results revealed that in spite of the satisfactory artificial damage induced in calcareous lithotypes by heating at 400 °C, the same weathering condition proved to cause no mechanical damaging in the quarzitic sandstone.

Based on the information provided above, the prior research is solely on the durability of the buildings. Still, there are very few studies estimating the time to reach a certain state of degradation, considering certain conditions of weathering to various environmental factors, as well as surface treatment, which plays an important role in protecting against stains of dirt on the building. In particular, the long-term durability of surface protective materials in outdoor service progress (under harsh environmental and pollution conditions, etc.) needs to be verified and evaluated further. 

This paper focuses on the analysis of the changes in the aesthetic properties of the surfaces and the water permeability of four types of coatings during xenon-arc light aging and further outlines to obtain more information on the degradation of the protective properties of the exterior wall protective coatings. Speeding up xenon-arc aging test used for weathering to xenon arc is the most damaging weathering factor with the high energy to break chemical bonds and start the degradation process. Observation of the appearance and water contact angle, color differences, gloss, roughness and water absorption, moisture permeability, and evaluation of the aesthetics between coatings and concrete were examined in order to monitor age-related changes. To assess the protective performance of coating materials, the resistance to soiling was determined. Therefore, the aim of this study is to investigate the relationships between coating deterioration and protection performance and also to show the influence of surface protection material on its physical and optical deterioration.

## 2. Materials and Methods

### 2.1. Materials

In this experiment, the materials were the white Portland cement with two types of mortar with w/c ratios of 0.45 and 0.55, as shown in Table 1. Because this is an evaluation of the surface protection material for the exterior precast concrete, the actual proportions of mortar of the precast concrete factory were used in this study as a reference. The cement had a specific gravity of 3.05 g/cm^3^. The fine aggregates had a density of 2.65 g/cm^3^ (crushed lime sand from Torigatayama, Kochi Prefecture, Japan).

### 2.2. Specimens

Mortar mixtures were prepared in the laboratory. One hundred specimens were produced with a size of 130 × 75 × 10 mm^3^ according to the corresponding JIS (Japanese Industrial Standard) methods. After casting and initial curing for 24 h in standard conditions, the specimens were removed from the molds and were cured in a controlled environment with a temperature of 20 °C and a relative humidity of 50% for 28 days. There were two types of mortar used in the experiment, with w/c ratios of 0.45 and 0.55. Fifty mortar specimens were used for 2500 h and fifty for 5000 h weathering, respectively.

### 2.3. Surface Treatments

After 28 days of curing, a brush was used to coat the one-sided surfaces of the mortar specimens with each type of selected surface coating. The concrete coating was applied following the application process recommended by the suppliers exactly. Four concrete surface protection materials were used in this study. Table 2 listed the technical information of surface protection materials. Uncoated mortar specimens were also prepared as control. After the surface coating, all specimens were cured in the dry condition at a temperature of 20 °C and a relative humidity of 30% for 14 days.

### 2.4. Accelerated Weathering (Weathering to Xenon-Arc Radiation)

Accelerated weathering of the surface coating materials on samples was performed for 5000 h in a xenon-arc radiation test chamber according to Japanese Standard JIS K 5600-7-7:2008 (ISO 11341:2004) [29]. The wetting and drying time were 1 cycle (120 min), with the wetting time of 18 min and the drying time of 102 min. Wetting condition: at temperature 38 ± 3 °C and relative humidity RH95% and the irradiance of 60 W/m^2^ (in the range of 300 nm~400 nm) water was sprayed on the surface. Drying condition: the irradiance of 60 W/m^2^ (in the range of 300 nm~400 nm), with the relative humidity RH of 50%, the black panel temperature of 63 ± 2 °C. The test specimens to evaluate each performance of coatings was carried out at different weathering hours, after 0 h (initial test), 2500 h and 5000 h, respectively. Table 3 lists the technical information about the accelerated aging test procedure.

### 2.5. Appearance Change

#### 2.5.1. Visual Observation

The visual observation was carried out according to the “Evaluation of Degradation of Coatings—Designation of the amount and size of defects and the intensity of uniform changes in appearance” defined in Japanese Standard JIS K 5600-8-1~6 [30]. The evaluation was carried out under the bright illumination that the defect or appearance change can be confirmed.

The indication of the degree of uniform changes in the coating surface, such as hue changes (e.g., yellowing) and calcification of the coating film, are shown in Table 4. If there were no other agreements between the receiving parties, the grade is indicated as an integer.

Three specimens for each treatment and three untreated specimens from each concrete were used for a total of thirty specimens tested.

#### 2.5.2. Color Differences

Color differences the visual characteristics test method according to Japanese Standard JIS K 5600-4-6 [31] were performed, and the results were evaluated using the color difference △E*_ab_ between two colors using the CIELAB color difference formula that is the geometric distance between two colors in the (CIE 1976) L*a*b* color space. 

The ΔE formulas are shown below:ΔE*_ab_ = [(ΔL*)^2^ + (Δa*)^2^ + (Δb*)^2^]^1/2^
ΔL* = L*_T_ − L*_R_
Δa* = a*_T_ − a*_R_
Δb* = b*_T_ − b*_R_
where L^∗^ is the lightness, a^∗^ is the green/red color component, b* is the yellow/blue color component. ΔL*, Δa* and Δb* indicate the differences in L*, a* and b* between two specimens. 

Each specimen was measured at four points (75 mm diameter) within the specimens (mean value was used).

#### 2.5.3. Gloss

Per JIS K 5600-4-7 [32], the glossiness was measured by “HORIBA IG-340 (HORIBA, Ltd, Kyoto, Japan)”. The axis of geometrically condition incidence light is typically at three angles of incidence 20° ± 0.5°, 60° ± 0.2°, and 85° ± 0.1°. The axis of the receiver shall match within ±0.1 with respect to the mirror image of the axis of incident light. 

A smooth surface of polished black glass or a mirror was placed at the sample position, so that the image of the light source is formed in the center of the viewing aperture of the receiver. The width of the irradiated part of the specimen shall be significantly larger than the surface structure to be estimated, so that an average over the whole surface is obtained. The generally accepted width value is 10 mm.

The measurement point is the same as the color difference.

#### 2.5.4. Roughness

Per JIS B 0601-2001 [33], the surface roughness was evaluated by the average roughness Ra (μm) according to “Mitutoyo SJ-210”(Mitutoyo Corporation, Kawasaki-shi, Kanagawa 213-8533, Japan). Measurement was performed by a roughness tester that meets the specifications of JIS B 0601-2001 “Geometric characteristic specifications (GPS)—surface properties of products: contour curve method—terms, definitions and surface properties parameters”. In both the x and y directions, four points were measured from a point 37.5 mm away from the angle toward the center.

#### 2.5.5. Contact Angle

The contact angle is a simple parameter to characterize the hydrophobicity of the surface-coated concrete. In order to evaluate the surface hydrophobic effect of the treatments, the static contact angle was measured at 10 different points of specimens treated on one face with a Drop Master 300 (Kyowa Interface Science Co., Ltd. Niiza-City, Saitama, Japan). A water drop of known volume was released on the concrete external surface by a flat needle placed from a known distance, determining the contact angle for the water drop pick-up.

### 2.6. Water Absorption Test

As water ingress into concrete is directly or indirectly responsible for all its degradation processes, the resistance after accelerated aging test of the treated and untreated specimens due to radiation and water ingress is the destructive element for these materials.

The water absorption tests were conducted with NSK Specification “Permeable Water Absorbent Preventive Material” determined using the Japan Standard NSKS-04 [34].

The prepared composite specimens were dried in oven at 80 °C until the specimen’s mass remained constant, followed by submerging in water for 24 h. The changes in the mass of the specimens used to evaluate the water absorption ratio was calculated by dividing the surface area (in cm^2^) with the weight growth (in mg) of the sample. For each treatment, three specimens of each concrete were used, and three untreated specimens were retained.

### 2.7. Moisture Permeability Test

Moisture permeability is the resistance of a material to water vapor diffusion through a unit of surface area. The moisture permeability test was performed on the specimens treated on the surface and on the untreated ones for comparison according to Japan Standard JSCE-K571-2005 [35]: specimens were weighed in surface-dry conditions after keeping immersed in water for 3 days. After that the container was weighed every day to determine the amount of water that exited the container after 7 days.

### 2.8. Anti-Soiling Test

The anti-soiling test method standards were devised in the previous report [36]. This research reproduced the flow-down pollution that occurs on the lower wall surface of the window, so that rainwater containing accumulated dust flows down uniformly to the sample surface. The contaminated water model is a carbon black (manufactured by FW 200: Orion Engineered Carbons) according to the method for pollution accelerated test for construction exterior wall materials by JSTM J 7602. The reason carbon black was chosen is because it easily disperses in water and stands out black by itself (Figure 1a).

We designed an experimental device to simulate carbon black liquid (Figure 1b). To imitate the traces left by raindrops, one streak stain was generated by dropping a 50 mL suspension onto a corrugated sheet at 2 drops/sec and letting it flow down from the corrugated sheet onto a test piece. After dropping, it was dried for 20 min. In order to verify the cleanability of the surface protective material, spray cleaning (Figure 1c) and ultrasonic cleaning (Figure 1d), two cleaning methods with different strengths, were used. 

In order to show the rain falling in the form of a curtain of water, the rain washing the carbon black trace was simulated using a water spraying machine. By spraying water on the wave board, water was allowed to flow down from the corrugated sheet to the test sample and washing was performed for 2 min. Ultrasonic cleaner, which generates a frequency of 40 kHz, is stronger than spray cleaning method. After 1 min of cleaning, 300 cc water was used to wash away the carbon black particles on the surface of the test specimens.

#### 2.8.1. Visual Observation

After dropping with the carbon black liquid, cleaning with a spray, and cleaning with ultrasonic waves, photographs were taken and the changes in appearance were visually observed.

#### 2.8.2. Color Difference

Brightness is the degree of brightness of a color, where L* = 0 represents black and L* = 100 represents white. The brightness difference was calculated by measuring the brightness of the entire surface of the test piece before and after the anti-soiling test using the image processing software ImageJ. The brightness difference, ΔL*, was determined from the following formula: ΔL* = |L* − L0*|, where ΔL* is brightness difference, L* is average brightness after carbon black liquid dropping test, and L0* is average brightness before carbon black liquid dropping test.

#### 2.8.3. Pollution Average Width

Pollution average width was measured by the image processing software ImageJ (ImageJ version 1.8.0_172). The threshold value was set to 150 after binarizing. The width within the length range of 15 cm from top to bottom was measured.

## 3. Results and Discussions

### 3.1. Visual Observation

Figure 2 shows the visual appearance of the uncoated and coated specimens after 0 h, 2500 h and 5000 h weathering of w/c 0.45 (a) and 0.55 (b). 

Regarding the differences among the surface coating materials, compared to the initial test (0 h data), the coating surface became rough on the silane1 A, silane2 B and silicate D surface specimens after 2500 h and 5000 h of weathering, indicating that the coating surface is deteriorated. The formation of roughing on the surface also has been reported in the literature on the surface of aged silicate and silane coatings, which is the effects of xenon-arc radiation weathering [37,38]. However, before and after 5000 h of weathering, C specimens, which were coated with fluor-resin coating, were intact. 

### 3.2. Brightness

Additionally, the study of the color differences of the surface coating materials was carried out before and after the xenon-arc light radiation. Figure 3 shows the brightness (L*).

All surface coating specimens gradually darkened after 2500 h and 5000 h weathering (except C). Regarding the differences among surface coating materials, there were practically no differences among coatings, since L* shows small values. This is agreement with the visual observation result in Figure 2. For both the 0.45 and 0.55 water cement ratios, the brightness shows identical tendencies but is slightly less pronounced in the case of the w/c 0.55 after 5000 h weathering.

### 3.3. Gloss

Gloss measurements of uncoated and coated specimens after initial 0 h, 2500 h and 5000 h of accelerated aging test are shown in Figure 4. 

Regarding sample gloss (measured at 60°), the results indicated an increase in gloss loss of all the specimens with increasing periods of xenon-arc light radiation, and after 2500 h of xenon-arc light radiation, gloss retention remained above 70% (except D). These gloss results are indicative of an increase in surface roughness, later confirmed by roughness measurements after 2500 h of xenon-arc radiation. Furthermore, over 60% of the initial gloss was lost from the silicate D specimen. The silicate D coating on the surface of the mortar specimens was deteriorated the same as the uncoated and un-weathered N specimens. 

Because D has lower contact angle value (<90°), which indicates that this result is due to the hydrophilicity of coating and it is not suitable for silicate surface impregnation treatment under the xenon-arc radiation environment). The particular reasons for this question will be explained through the reaction mechanism and morphological of silicate in future work.

### 3.4. Roughness

Results of roughness are shown in Figure 5. During weathering, the original surface of the mortar samples appeared gradually due to the destruction of the coating. After 2500 h weathering, the entire original surface of the mortar appeared clearly (except C). After weathering 5000 h (except C) roughness increased by over a factor of 22 mm ± 2 mm, hence, exposed concrete surface was observed. In the case of the C specimens no roughness was observed.

As shown in Figure 2 the initial surfaces of all of the specimens were very smooth and free of any signs of cracking or scratches. After 2500 h of xenon-arc radiation weathering, small pockmark-like features formed on the N, A, B, and D specimens that showed the direct evidence of top-coat loss from the basecoat, via the presence of large cavities and pits on the remaining surface. Fluor-resin C showed almost no loss of roughness, indicating that fluor-resin coating has a good weather resistance. The reason why the fluor-resin coating did not show degradation is due to the helical distribution of fluorine atoms along the carbon chains, and the relatively close interaction between the fluorine atoms and the carbon chains lead to a shielding effect which imparts good UV resistances [39]. Wood et al. [40] and Sung et al. [41] also reported the fluorine atoms could form a continuous uniform coating film to develop a structure with strong chemical bonding, which makes it difficult to produce the free radicals which cause coating decomposition.

### 3.5. Contact Angle

The water contact angle is commonly used to characterize a wetting property of a surface. The hydrophobic properties of these coatings can be quantified using the value of contact angle which is greater than 90°. The lower contact angle value (<90°) indicates the hydrophilicity of coating.

Figure 6 shows that on average all of the coating specimens had different contact angles before xenon-arc radiation weathering (time zero). It then increased further over time, reaching 42° after 5000 h weathering, and the D coating specimen had same tendency with N, with a lower contact angle value (<90°) indicating the hydrophilicity of the coating. The contact angles for silane1 A, silane2 B were from 78° to 38°, and from 130° to 30°, respectively. After xenon-arc radiation weathering, weathering decreased the contact angle significantly, as was the case with N after 2500 h. This indicates that the hydrophobic behavior of the coated specimens decreased almost the same as the uncoated ones. Regarding the visual observation of roughness after 2500 h and 5000 h in Figure 3; Figure 5, the coating surface appeared rough on all of the specimens because of weathering due to xenon-arc light. These surface defects could explain the loss of transmittance and contact angle. The results of contact angle obtained for C coating specimens showed relatively stable behavior. Even though C had some fluctuations during the 5000 h weathering, the contact angles still remained at around 80°. In conclusion, the most resistant coatings to this accelerated aging test were the C coatings.

### 3.6. Water Absorption Test 

Figure 7 shows the results of the water absorption test for mortar (a) and (b). It can be seen that the uncoated sample (N) quickly absorbs water before weathering. After about 2500 h of weathering, the water uptake decreased; after 5000 h there was increased water absorption. Specimen D coated with silicate pore blockers showed a similar tendency. This suggests that this is probably due to the hydrophilic nature of the deposited material [42]. Specimens coated with silane A with high penetration and silane B with high water repellency exhibited similar water absorption properties, with hardly any water absorption during the whole weathering period, which, together with the contact angle results in Figure 6, suggests their hydrophobicity was still kept after the xenon-arc radiation weathering, because of the deep penetration of the coatings. Although there was a slight decrease in roughness and brightness (L*) after 5000 h of xenon-arc radiation weathering (Figure 2, Figure 3 and Figure 5), C specimens showed reduced water absorption after 5000 h weathering. This result is because the fluorine atoms can form the continuous uniform coating film to develop the structure with strong chemical bonding and make it difficult to produce the free radicals which cause coating decomposition [40,41]. The particular reasons for this can be explained through the test method by transmission infrared spectroscopy (FT-IR) [43] and nuclear magnetic resonance spectroscopy (Si NMR) [44], and the surface of the fluor-resin coating was analyzed using reflection infrared spectroscopy (RA-IR) and X-ray spectrometer (EDS) [45].

### 3.7. Moisture Permeability Test

The moisture barrier properties of the silane, silicate and fluor-resin coatings with mortar was studied using the moisture permeability test. The weight moisture permeability over UV weathering time for mortar with and without coatings are shown in Figure 8.

The mortar specimens with and without coatings were fully saturated after being immersed in water 3 days before the moisture permeability testing in the constant temperature and humidity room. Therefore, their weight moisture permeability represents the specimen moisture release weight at 7 days.

After 0 h xenon-arc radiation weathering, it is obvious that the presence of silane A and B coatings significantly reduced the moisture permeability of mortar. The penetration depth of silane in the concrete by the coating along with the excellent barrier characteristics of silane were mainly responsible for the reduced moisture permeability. After 5000 h xenon-arc radiation weathering the amount of moisture released was basically the same. This observation implies that all of the coatings were destroyed.

As expected, a higher penetration was observed at a mortar w/c ratio of 0.55 compared to a mortar w/c ratio of 0.45, which is due to the higher porosity of the former type of mortar.

### 3.8. Anti-Soiling Test

Anti-fouling is a surface property that prevents dirt from sticking to a surface. In order to study the anti-soiling ability of a coating, anti-soiling efficiency of aged coatings after 5000 h xenon-arc radiation weathering was analyzed. 

#### 3.8.1. Visual Observation

The pollution resistance capability of the four types of coatings was checked by visual observation. Carbon black raindrops as contaminant were allowed to flow down on the specimens as shown in Figure 9(a1,b1). 

Initially, concerning the transmittance results, specimens with the uncoated N and coatings D showed almost the same width of carbon black raindrops on surface. The was probably due to the hydrophilic nature of the N and D.

Specimens with the coatings A, B and C showed that the carbon black liquid in a spherical drop ran down the surface along preferential paths, and the carbon black liquid raindrops were narrower than the uncoated N on the surface, especially B. Similar observations have been made by Charola et al. (2008) [46], on marble statues. The hydrophilic properties of the treated surface protect rainwater from distributing lead to the formation of spherical drops along the surface.

However, surface coatings tend to degrade over time under weathering due to xenon-arc light radiation. In all coating specimens the carbon black raindrops became as wide as in the uncoated N specimens after 2500 h and 5000 h aging, except C. Together with the result of visual observation photograph and roughness, contact angle had same tendency in Figure 3, Figure 5 and Figure 6 along with the water absorption results in Figure 7, suggesting their hydrophobicity was still kept after the aging test because of the deep penetration of the coatings. Some studies indicate that the residual effect of such coatings still gives better service life for treated concrete [7,22]. 

Afterwards, to check the cleanability of the surface protection material, spray cleaning and ultrasonic cleaning, two washing methods with different strengths, were used on the contaminated surface, respectively (Figure 9(a2,a3,b2,b3)). It can be observed that the carbon black raindrops were not easily removed by spray cleaning, as shown in (Figure 9(a2,b2)). However, the carbon black vestiges were easily removed by ultrasonic cleaning, as shown in (Figure 9(a3,b3)). This suggests that the four coatings can be removed the pollution, but strength is needed in the washing process to achieve this cleaning.

#### 3.8.2. Pollution Average Width

After cleaning, pollution average width was tested to observe possible cleanability of the four types of surface protective material. 

As can be seen in Figure 10a,b, both w/c ratios of 0.45 and 0.55 specimen measurements exhibited changes in pollution average width in initial test (not weathering), which after spray cleaning, and ultrasonic cleaning. Even though the spray cleaning method was lower than ultrasonic cleaning method of all of specimens (except coatings D), a significantly lower value was achieved compared with uncoated N specimens. This means that the four types of coating materials have cleanability. Meanwhile, the coatings for specimens A and B exhibited similar anti-soiling features to the C coated specimens. The most hydrophilic coated specimen D was confirmed to be anti-soiling as the pollution average width was on its surface was small. 

This can be ascribed to the higher hydrophilicity of silicate-based coatings, and to the actual onset of cleaning based on high hydrophilicity, which allows better removal by water of the dirt accumulated on the surface [47,48,49]. 

However, after washing, it was observed that the pollution average width of all of the specimens after weathered for 5000 h were the same as or much smaller than the initial value. These results were consistent with those reported previously [50]. Physical degradation may also induce an apparent self-cleaning behavior when pristine surface is exposed after the detachment of aged portions.

#### 3.8.3. Brightness Difference

As seen in Figure 11, after the 5000 h accelerated aging test, all of the specimen measurements exhibited large changes color after weathering, with decreased ΔL*. 

Overall, these findings indicated that after 5000 h of xenon-arc light weathering, accumulated dirt led to the greatest change in color (i.e., appearance) but these effects were easily remedied by sample washing.

## 4. Conclusions

In this work, four kinds of external wall protective coatings were used to investigate the effect of the artificial accelerating test, which was performed to evaluate the aesthetics as well as the permeability resistance of the surface protective materials to the degradation of external wall protective coatings due to xenon-arc light radiation. Two main conclusions can be drawn. 

First, the permeability of the four coats was assessed. Static contact angle and surface water absorption analyses were carried out. From the results, a decrease in the contact angle was observed after the aging; in particular, the silicate coating became highly hydrophilic after 5000 h of aging. The contact angle of the fluor-resin coating was substantially not influenced after 5000 h of aging. On the other hand, xenon-arc radiation reduced surface water absorption significantly on treated specimens, while it did not affect untreated ones. In this way, the silane type and fluor-resin type coatings do not seem to bring greater water absorption, a potential source of damage to concrete surfaces.

Second, the anti-fouling-cleaning ability of the four coats was then assessed. Specimen measurements exhibited changes in pollution, average width after weathering, increased spray cleaning, and ultrasonic cleaning. However, after washing, it was observed that the pollution average width of all the specimens after weathered at 5000 h were nearly the same or smaller than the initial value. Initially, the surface could also be observed, for the paths of carbon black liquid were narrow for the A and B specimens treated with a water-repellent. Initially, the water ran down the surface along preferential paths and spherical drops were noticed on the surface. For the 2500 h test, however, the moistness of the surface was increased, with a loss of the water-repellent effects after the 2500 h and 5000 h tests for silane types coated specimens A and B. Specimen C, which was coated with fluor-resin, kept its water-repellent properties substantially longer.

## Figures and Tables

**Figure 1 materials-14-06896-f001:**
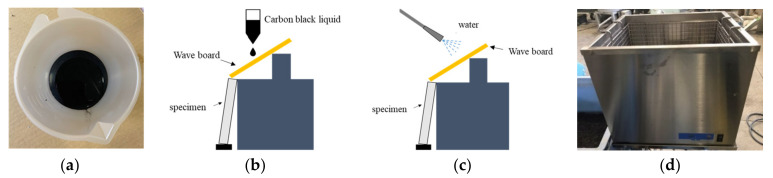
The carbon black liquid dropping method (**a**); carbon black liquid (**b**); the spray cleaning method (**c**); ultrasonic cleaning machine (**d**).

**Figure 2 materials-14-06896-f002:**
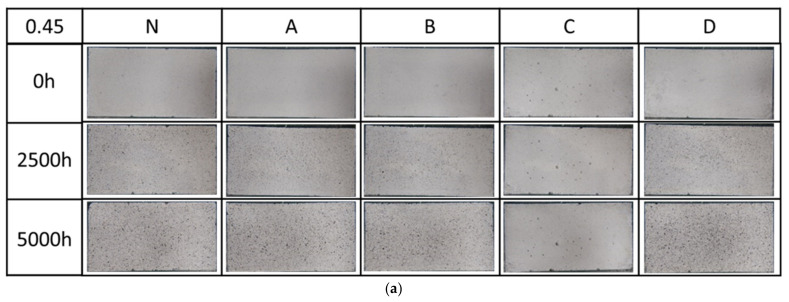
The Visual observation of the uncoated and coated specimens after 0 h, 2500 h and 5000 h weathering of w/c 0.45 (**a**) and 0.55 (**b**).

**Figure 3 materials-14-06896-f003:**
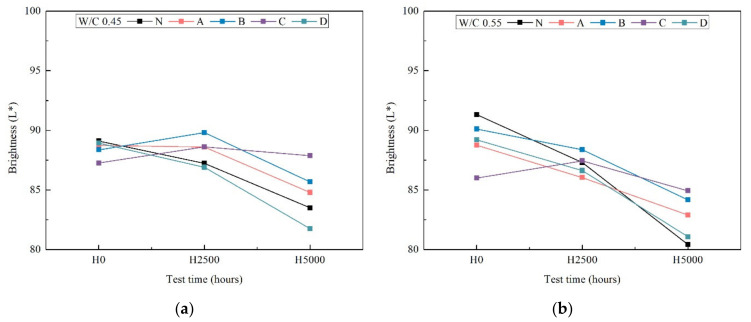
The brightness (L*) of color difference after accelerated aging test after 0 h, 2500 h and 5000 h weathering of w/c ratio 0.45 (**a**) and 0.55 (**b**).

**Figure 4 materials-14-06896-f004:**
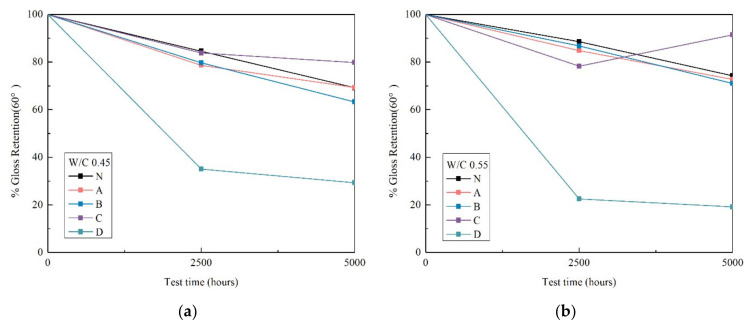
Gloss results of uncoated and coated specimens after 0 h, 2500 h and 5000 h weathering of w/c ratio 0.45 (**a**) and 0.55 (**b**).

**Figure 5 materials-14-06896-f005:**
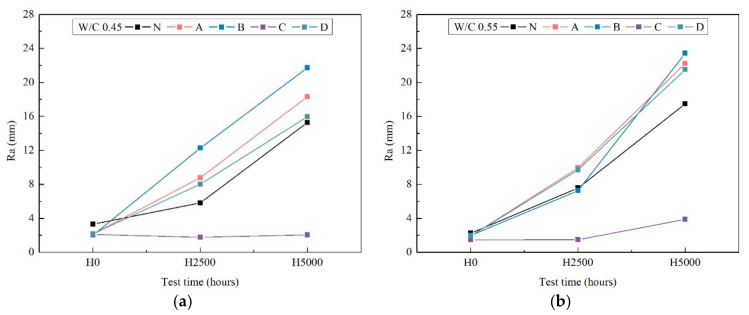
Roughness of surface coating after 0 h, 2500 h and 5000 h weathering of w/c ratio 0.45 (**a**) and 0.55(**b**).

**Figure 6 materials-14-06896-f006:**
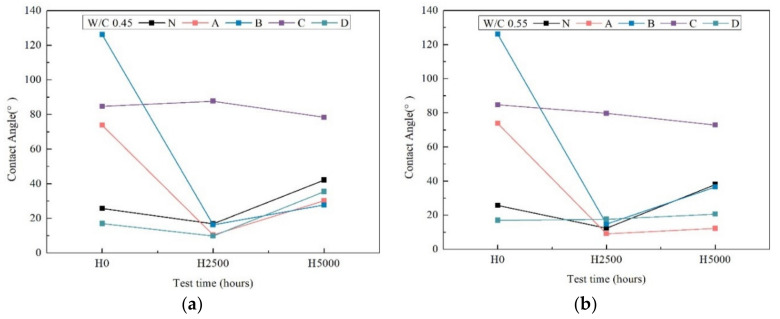
Contact angle test after 0 h, 2500 h and 5000 h weathering of w/c ratio 0.45 (**a**) and 0.55 (**b**).

**Figure 7 materials-14-06896-f007:**
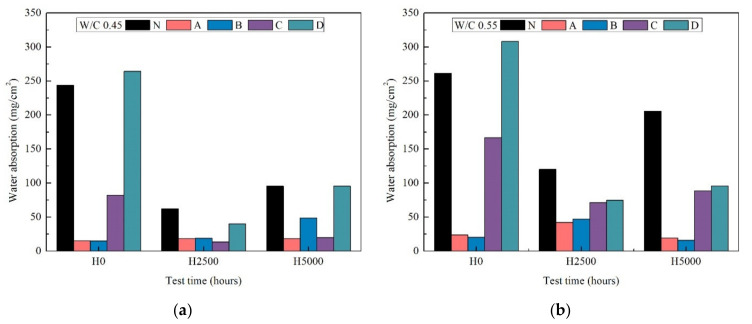
Water absorption test according to Japan Standard NSKS-04 on uncoated and coated specimens of mortar at w/c ratios 0.45 (**a**) and 0.55 (**b**).

**Figure 8 materials-14-06896-f008:**
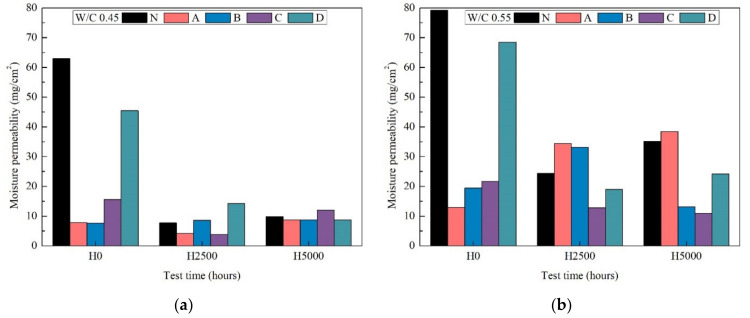
The moisture permeability test after 0 h, 2500 h and 5000 h weathering of w/c ratio 0.45 (**a**) and 0.55(**b**).

**Figure 9 materials-14-06896-f009:**
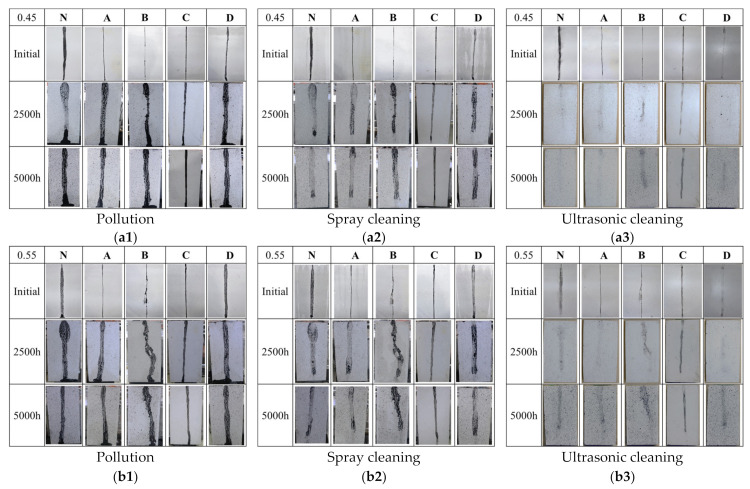
The Visual Observation of the uncoated and coated specimens after pollution (**a1**,**b1**), spray cleaning (**a2**,**b2**) and ultrasonic cleaning (**a3**,**b3**), w/c ratio 0.45 (**a1**,**a2**,**a3**), 0.55 (**b1**,**b2**,**b3**) of accelerated soiling tests.

**Figure 10 materials-14-06896-f010:**
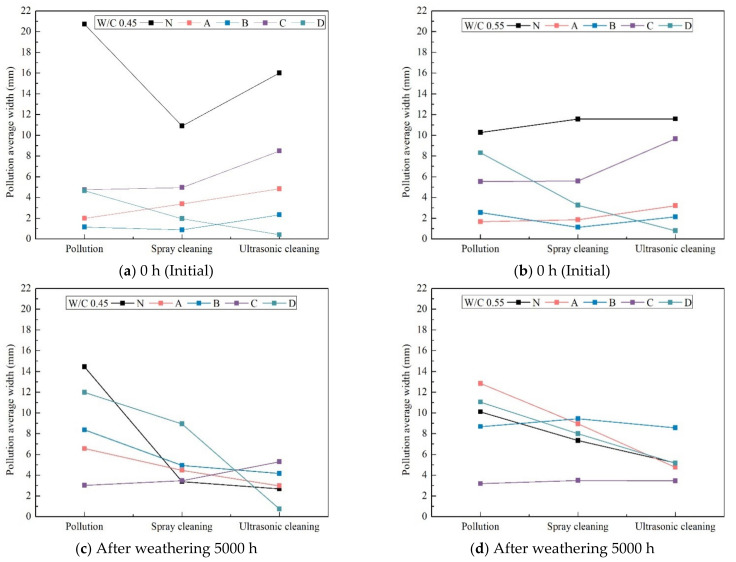
Pollution average width measurements of all protective coatings at different times of xenon-arc light exposure after pollution and cleaning (spray and ultrasonic cleaning).

**Figure 11 materials-14-06896-f011:**
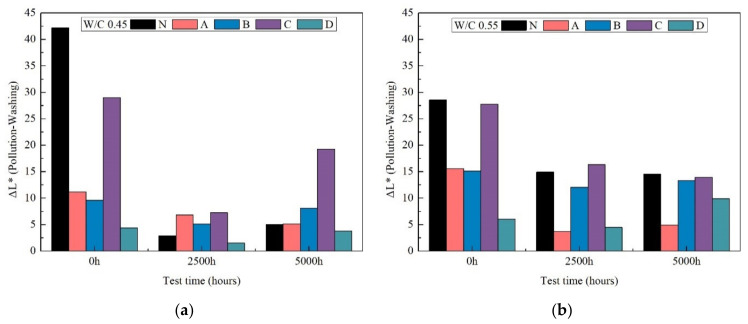
The brightness difference after accelerated fouling tests of 0 h, 2500 h and 5000 h weathering for w/c ratio 0.45 (**a**) and 0.55 (**b**).

**Table 1 materials-14-06896-t001:** Mix proportions of mortar.

Water Cement Ratio	Cement	Flow	Air Content	Material Content (kg/m^3^)		
w/c		(mm)	(%)	W	C	S
0.45	white Portland cement	200	3.0	228	415	1606
0.55				228	508	1525

**Table 2 materials-14-06896-t002:** Surface protective materials.

Types and Symbol	N		A		B		C		D	
	Un-Coated		Silane1		Silane2		Fluor-Resin		Silicate	
**Coating procedure**	First coating	-	-	-	-	-	water-based water repellent	200 g/m^2^	-	-
	Curing time	-	-	-	-	-	More than 16 h		-	-
	Medium coating	-	-	-	-	-	Fluor-resin	100 g/m^2^	-	-
	Curing time	-	-	-	-	-	More than 3 h		-	-
	Top coating	-	Silane	200 g/m^2^	Silane	200 g/m^2^	Fluor-resin	100 g/m^2^	Aqueous silicate	250 g/m^2^
	Curing time	-	More than 4 h		More than 6 h		More than 24 h		24 h	

**Table 3 materials-14-06896-t003:** The method of accelerated aging test.

	Time	Temperature	Black Panel Temperature	Relative Humidity	Irradiance
	min	°C	°C	%	(W/m^2^)
**Wetting**	18	38	-	95	60
**Drying**	102	-	63 ± 2	50 (40–60)	60

**Table 4 materials-14-06896-t004:** Displaying grade of appearance change.

Grade	Appearance Change
0	No change (i.e., there is no change to be observed.)
1	Very small (i.e., barely observed change.)
2	A slight (i.e., a clearly observed change.)
3	Medium (i.e., a very clearly observed change.)
4	Critical (i.e., substantial change.)
5	Very marked change

## Data Availability

The data presented in this study are available on request from the corresponding author.

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
