# Peer review of "Investigation on Aesthetic and Water Permeability of Surface Protective Material under Accelerated Weathering"

_materials, 2021, doi:10.3390/ma14226896_

Round 1

Reviewer 1 Report

The paper describes investigations into physical changes of protective surface systems for concrete facades under weathering impacts. The structure of the paper is appropriate, the methods are sufficiently described. Results and  conclusions are are comprehensible. Language is the weakest point of the manuscript, it is often hard to follow the author's thoughts.

Some editorial and content-related notes:

  1. Line 124 ff: How many specimens have been weathered and tested for each protection system? This is not clear.
  2. Line 151: What exactly is the wet state? Is spraying used? Or condensation?
  3. Line 152: The chosen weathering procedure for sure does not correlate to 209 days as radiation was turned on for all 5000 hrs. of testing.
  4. General remark: Is radiation or water ingress the main destructive element for these materials/surfaces? Or the combination? In the text, sometimes it reads like water is the main stressor (e.g. line 187), on the other hand oftentimes only radiation is mentioned or pointed out (e.g. lines 306, 316, 349 etc.).
  5. Line 237: "antifouling property test", this term was not used before. What excatly is meant?
  6. Line 249: The sentence makes no sense in this way.
  7. Check references to Figures 2 and 3. Seems to be odd sometimes.
  8. Line 266: "...with initial rapid change after 2500 hours...". Does not fit to quite some of the paths which show an increase of brightness after 2500 hrs.
  9. Line 288: sentence completely unclear.
  10. Fig. 10: Please explain the differences between a, b, c and d in the Figure caption and text. Does not become clear.
  11. Line 426: Not four kinds of coatings?
  12. Line 446: 2500 hours test week? This amount of hours does not fit to a week.

Reviewer 2 Report

Dear Authors, the paper could be of some interest but some major revisions are needed. In the introduction, some more discussion on general artificial weathering is needed. It is not just applied on coatings but also on other materials exposed to weather. Thus a general discussion as introduction is needed in order to permit the reader to understand that this is a generalized problem. And you apply your research on a specific issue of a general problem.
Thus the reader would benefit from a discussion including other applications like (e.g.):
Polymers http://dx.doi.org/10.1016/j.compositesa.2017.01.016
wood 10.1007/s00107-014-0791-y
stones https://doi.org/10.1016/j.culher.2012.11.026
etc
So please do include these papers and briefly discuss them

Section 2.5 you used JSK standards. Unfortunately they are not international. Please briefly describe the methodology so as to explain the reader the approach
section 2.5.2 why investigating the color? which scientific information does the color provide? only aesthetic?
section 3.3 The D sample is very different from the other ones. Anyway an full explanation is missing. Please add
line 292 Ok I get from the figure that specimen C ha no roughness. But why? same as the previous comment

Reviewer 3 Report

The authors present an interesting paper related to the investigation of aesthetic and water permeability of surface protective material under accelerated weathering. The paper is well-written and well-structured, and the arguments are clearly developed.

Author Response

Thank you for your review.

Reviewer 4 Report

This paper describes aesthetic and water permeability of surface protective material under accelerated weathering. The subject of this manuscript is relevant to the Materials. I would encourage the authors to include more discussion of their data.  The other shortcomings in this manuscript were presented as following.
1. The novelty of the paper should be stressed: what is really new with respect to the state of the art. As authors themselves highlight, the use of silane, fluor-resin silicate as surface protection materials has been the subject of intensive studies is a common practice worldwide.

  1. What is the shortage of the knowledge in the literature? You should address it in Introduction. The authors should provide a citation for each of these supposedly outstanding properties. Similar citations in [7–13] and [14–21] should be given more information for each of the reference.
  2.  The description of the choice of the w/c ratios of 0.45 and 0.55 is not clear. Please provide the reason.
  3.  It illustrates that it is not suitable for silicate surface impregnation treatment under high humidity condition. The particular reasons for this question will be explained through the reaction mechanism and morphological of silicate in the future work. Please provide some theories or compare the results with previous researches.

5. Specimens C even showed a reduced water absorption after 5000 hours weathering. Because the hydrophobic impregnation prevents the penetration of additional external water. Please provide some theories or compare the results with previous researches.

Round 2

Reviewer 2 Report

Dear Authors, thank you for your revised version

Author Response

Thank you for your review.